# Clustering for Regional Time Trend in the Nonstationary Extreme Distribution

**Sungchul Hong [1], Jong-June Jeon [1],* and Yongdai Kim [2]**

[1]  Department of Statistics, University of Seoul, Seoul 02504, Korea; hsc930526@gmail.com
[2]  Department of Statistics, Seoul National University, Seoul 02504, Korea; ydkim0903@snu.ac.kr
*   Correspondence: jj.jeon@uos.ac.kr

**Abstract:** Since the estimation of tail properties requires a stationarity of observations, it is necessary to develop a de-trending method not dependent on underlying distributions for nonstationary hydrological processes. Moreover, de-trending has been independently applied to hydrological processes, even though the processes are observed in geometrically adjacent sites. This paper presents a distribution-free de-trending method for nonstationary hydrological processes. Our method also provides clustered regional trends obtained by sparse regularization in a general distribution. It aggregates the parameter estimation and clustering within a unified framework. In the simulation study, our proposed method has superiority over other compared methods with respect to MSE and variance of coefficients. In real data analysis, the clustered trends of the annual maximum precipitation in the South Korean peninsula are reported, and the patterns of the estimated trends are visualized.

**Keywords:** clustering; fused lasso; nonstationary distribution; regional frequency analysis; time trend estimation





## 1. Introduction

One of the main problems in extreme data analysis is the lack of observations and the scarcity of data, leading to large variances of the estimators, which deteriorate the predictive performances of the statistical model [1–4]. The regional frequency analysis (RFA) (e.g., [1,2]) has been commonly employed to overcome this problem. The RFA assumes that the hydrometeorological variables across homogeneous regions follow a common probability distribution. The observations are pooled on the identified homogenous regions, and the parameters of the distributions are estimated by the pooled observations. Therefore, the regionalization of the analyzed basins is a prerequisite for the RFA procedure, and its success mainly depends on the selection of appropriate clustering methodology.

Various clustering techniques have been developed and applied in RFA [1–8]. The geomorphological variables (e.g., drainage area, basin elevation, soil runoff coefficient) and meteorological variables (e.g., mean precipitation, quantiles of precipitations, annual mean degree days over 0 °C) are used to cluster the homogeneous regions. There is no consensus as to which variables should be considered in clustering. Rather, the choice of variables and clustering technique depends on the purpose of the analysis. However, since a specific class of distributions is determined after the identification of homogeneous regions, it is common to use the distribution-free clustering method.

A few researchers [9–13] proposed a new RFA under nonstationary circumstances. The main problem of the conventional RFA is its asymptotical validity based on the stationary assumption. For example, the L-moments used in RFA [1,2] should be obtained by the order statistics from a stationary distribution. When the underlying distribution is nonstationary, the large sample theory is not valid anymore. Furthermore, the homogeneity of the distributions is not clearly defined under nonstationary circumstances, so there is no universal framework to characterize regional homogeneity in the RFA.

Ref. [11] assumes the nonstationary distribution with a linear trend and proposes the nonstationary RFA where homogeneity is defined over de-trended distributions. According to Ref [11], considering the homogeneous regions improves the estimation of nonstationarity, which affects the final decision of the RFA. Thus, we propose a distribution-free de-trending method for nonstationary hydrological processes, which aggregates the parameter estimation and clustering within a unified framework. Building on Ref [11], a two-stage de-trending method is considered: the decomposition of regional homogeneities into stationarity and the estimation of nonstationarity. The first one includes regional specific factors considered in the conventional RFA, and the second one includes the patterns of changes in hydrometeorological variables.

In this study, it is assumed that the patterns are represented by linear functions of time, which implies nonstationarity, and we focus on detecting homogeneities on the linear trends. This study proposes a new methodology to cluster the linear trends with use of sparse regression models [14–20]. The proposed method simultaneously facilitates detection of potential nonstationarity and estimation of the regression parameters without any assumption of an underlying distribution. That is, the proposed method is a distribution-free clustering technique for grouping nonstationarity.

This paper is organized as follows. Section 2 introduces the regression model to estimate linear time trend of a nonstationary distribution. Section 3 explains the regularization method for the regression model and defines the penalized risk function for estimating the linear trends in the distribution. In addition, the optimization and the model selection follow in this section. Section 4 illustrates some numerical results by grouping coefficients and conducts a real data analysis. Discussion and concluding remarks are followed in Section 5.

## 2. Nonstationary Distribution with a Linear Time-Variant Mean

In the RFA, the class of distributions is identified only after determining the homogeneous regions. Because the regionalization contains the identification of the characteristics in distributions, it is not recommended to pre-determine the class of distributions before the regionalization. To avoid the misspecification of an underlying distribution, it is assumed that $U_{tj} = \beta_j t + \epsilon_{tj}$, where $\beta_j$ is a trend parameter at site $j$ and $\epsilon_{tj} \sim_{iid} f_j$ for all $t$. Here, $f_j$ denotes a general probability density function (pdf) at a site $j$. For example, $f_j$ is the pdf of a three-parameter distribution with the location $(\xi_j)$, scale $(\alpha_j)$ and shape $(\kappa_j)$ parameters, such as the GEV distribution. The homogeneity on the distribution $f_j$s is investigated by L-moments in the conventional RFA. Apart from the homogeneity of the distributions $f_j$s, the homogeneity in the linear time trend is also defined by the values of $\beta_j$s. Thus, two types of homogeneity in stationary and nonstationary parts are considered in the model, and estimation of the latter one is of our interest in this study.

Let $Y_j$ be the time domain of the site $j$, and let the true parameter of linear trend coefficients at a site $j$ in Equation (1) be $\beta_j^*$. Then, the distribution of $\widetilde{u}_{tt'j} = u_{tj} - u_{t'j}$ is symmetric at $(t - t')\beta_j^*$ for all $t$ and $t'$. That is, it can be rewritten by $\widetilde{u}_{tt'j} = (t - t')\beta_j^* + \widetilde{\epsilon}_{tt'j}$ with $\widetilde{\epsilon}_{tt'j} = \epsilon_{tj} - \epsilon_{t'j}$ of which the distribution is symmetric at zero. If $\beta_j^* = \beta^*$ for all $j$, or if there is a global mean trend in considered sites, then the $\beta^*$ can be estimated by $\hat{\beta} = (\hat{\beta}_1, \dots, \hat{\beta}_p)$, which is minimized. Let $\widetilde{Y}_j = \{(t, t') : t, t' \in Y_j, 0 < |t - t'| \leq m_j\}$ for some $m_j$, $j = 1, \dots, p$, then the objective function is given by

$$\sum_{j=1}^{p} \sum_{(t,t') \in \widetilde{Y}_j} \left| \widetilde{u}_{tt'j} - (t - t')\beta_j \right|, \tag{1}$$

subject to $\beta_j = \beta_{j'}$ for all $1 \leq j, j' \leq p$. When $m_j$ is the number of elements of $Y_j$, Equation (1) is known as rank-based regression model, and $\hat{\beta}$ is a generalization for the classical Wilcoxon–Mann–Whitney rank statistics for independent observations [11,21]. In addition, $\hat{\beta}$ can be understood as the least absolute deviation (LAD) estimator, which is an M-estimator for minimizing the empirical risk with $l_1$ loss function. Its asymptotic properties are well studied by Refs [22,23].

The estimation method of minimizing the empirical risk function with $l_1$ loss function can be applied to analyze the linear trends combining the regionalization method. Assume that the time trends are clustered with $q$ groups. Let $\boldsymbol{\beta} = (\beta_1, \ldots, \beta_p) \in R^p$ and $\boldsymbol{U}(\boldsymbol{\beta}) = \{\gamma_1, \ldots, \gamma_q\}$ be the set of the unique values of $\boldsymbol{\beta}$. Additionally, let $\boldsymbol{G}(\boldsymbol{\beta}) = \{G_1(\boldsymbol{\beta}), \ldots, G_q(\boldsymbol{\beta})\}$ be the set of indices of grouped variables with $\beta_j = \gamma_k$ for all $j \in G_k(\boldsymbol{\beta})$. For example, let $p = 4$ and $\beta_1 = 1$, $\beta_2 = -1$, $\beta_3 = 1$, $\beta_4 = -1$. Then, set $q = 2$, $\gamma_1 = 1$, $\gamma_2 = -1$, and $G_1(\boldsymbol{\beta}) = \{1,3\}$, $G_2(\boldsymbol{\beta}) = \{2,4\}$ such that $\boldsymbol{U}(\boldsymbol{\beta}) = \{1, -1\}$ and $\boldsymbol{G}(\boldsymbol{\beta}) = \{\{1,3\}, \{2,4\}\}$. When ideally $\boldsymbol{G}(\boldsymbol{\beta}^*)$, the true cluster, is known, the parameter $\boldsymbol{\beta}$ can be estimated by

$$\hat{\boldsymbol{\beta}} = argmin_{\boldsymbol{\beta} = (\beta_1, \ldots, \beta_p)} \sum_{j=1}^{p} \sum_{(t,t') \in \widetilde{Y}_j} \left| \widetilde{u}_{tt'j} - (t - t')\beta_j \right|, \tag{2}$$

subject to $\beta_j = \beta_{j'}$ for all $j, j' \in G_k(\boldsymbol{\beta}^*)$ $(k = 1, \ldots, q)$.

By the constraint of Equation (2), each estimate in $G_k(\boldsymbol{\beta}^*)$ for $k = 1, \ldots, q$ has the same value. However, $G_k(\boldsymbol{\beta}^*)$ is unknown in practice, such that such $\hat{\boldsymbol{\beta}}$ is an ideal estimate in Equation (2). If the regions are clustered with the same linear time trend, the estimation of the coefficient $\beta_j$ is expected to be improved by pooling data. Otherwise, the improvement is not expected due to poor clusters of regions. That is, when the pooled observations for regionalization are used, the bias of the estimator can be caused by heterogeneity of populations with wrong clusters. That is, the clustering regions play a key role of regionalization. In the next section, we explain a new method to estimate the clusters and the linear time trend simultaneously.

**Remark 1.** *It can be assumed that $U_{tj} = h_j(t) + \epsilon_{tj}$, where $\epsilon_{tj} \sim_{iid} f_j$, where $h_j$ is a time-varying locational function. In this case, $\widetilde{U}_{tt'j} = U_{tj} - U_{t'j}$ is also symmetric at $h_j(t) - h_j(t')$, and the nonparametric method [24] can be employed to estimate $h_j$.*

## 3. Proposed Method

The proposed method is based on a regularization method in the regression model, which is widely studied in statistics and computer science. First, candidates of clusters are obtained by the regularization method [16]. Second, sets of coefficients are recovered by applying the method to estimate the linear time trend. At last, the estimated coefficients are selected according to Bayesian information criterion [25].

### 3.1. Regularization Method for Regression Models

First, the empirical risk function Equation (2) is reformulated as a typical regression model with predictors. Let $l_j$ be the number of elements in the set $\widetilde{Y}_j$ for $j = 1, \ldots, p$ and $r_j : \widetilde{Y}_j \mapsto \{1, \ldots, l_j\}$ be a bijective function, which denotes the index of each element $(t, t') \in \widetilde{Y}_j$. Let $\widetilde{x}_{sj} = t - t'$ and $\widetilde{y}_{sj} = \widetilde{u}_{tt'j}$ for $s = r_j(t, t')$, and let $\widetilde{x}_j = \left( \widetilde{x}_{1j}, \ldots, \widetilde{x}_{l_jj} \right)^T \in R^{l_j}$ and $\widetilde{y}_j = \left( \widetilde{y}_{1j}, \ldots, \widetilde{y}_{l_jj} \right)^T \in R^{l_j}$ be the predictor and response vector in Equation (2). By concatenating the $\widetilde{x}_j$s and the $\widetilde{y}_j$s, extended vectors $X_j = \left( 0_{s_{j-1}}^T, \widetilde{x}_j^T, 0_{n-s_j}^T \right)^T$ and $y = \left( \widetilde{y}_1^T, \ldots, \widetilde{y}_p^T \right)^T$ are defined, where $0_p$ is the $p$-dimensional zero valued column vector and $s_j = \sum_{k=1}^{j} l_k$, and $n = s_p$. Finally, let $\mathbf{X} = (X_1 \cdots X_p)$ be $n \times p$ design matrix in the regression model and $\mathbf{x}_i$ be the $i$th row vector of $\mathbf{X}$, and let $y_i$ be the $i$th element of $\mathbf{y}$. Then, the empirical risk function Equation (2) is written by the terms of $y_i$ and $\mathbf{x}_i$ as $L(\boldsymbol{\beta}) = \sum_{i=1}^{n} |y_i - \mathbf{x}_i^T \boldsymbol{\beta}|$, which is used in estimating the coefficients in the LAD regression model. The regression coefficient is estimated by minimizing $L(\boldsymbol{\beta})$, and the regularization method is applied to the $L(\boldsymbol{\beta})$.

The regularization method gives a sparse estimate whose values are exactly zero or grouped. The regularized estimator is defined by the minimizer of $L(\boldsymbol{\beta}) + \mathrm{p}_\lambda(\boldsymbol{\beta})$, where $\mathrm{p}_\lambda(\cdot)$ is a non-negative valued function depending on $\lambda \geq 0$. When $\mathrm{p}_\lambda(\boldsymbol{\beta}) = \lambda \sum_{j=1}^p |\beta_j|$, $\mathrm{p}_\lambda(\boldsymbol{\beta})$ is called the lasso penalty function [26], which gives a shrinkage estimate toward zero, such that some components in the estimate become exactly zero. Using this property, a sparse estimator, simultaneously achieving model selection and parameter estimation, is obtained. In our example, the model automatically detects the site with no linear trend. There is another penalty function called the fused lasso penalty, defined by $\mathrm{p}_\lambda(\boldsymbol{\beta}) = \lambda \sum_{(j,k)\in N} |\beta_j - \beta_k|$ with $N \subset \{(j,k) : 1 \leq j, k \leq p\}$. The fused lasso penalty produces a shrinkage estimate toward centers of the neighborhoods pre-defined by the network $N$. By a proper design of the network $N$, the fused lasso penalty is used for detecting change point of time series [27,28] and grouping regression coefficient [16,19] in the regression model. A regularized method using this fused lasso penalty in the regression model is proposed, as follows. In our case, the estimator is given by

$$\hat{\boldsymbol{\beta}}_{\lambda_1, \lambda_2} = L(\boldsymbol{\beta}) + \lambda_1 \sum_{j=1}^p |\beta_j| + \lambda_2 \sum_{(j,k)\in N} |\beta_j - \beta_k|, \tag{3}$$

where $N$ is a set of index pairs of sites, and $\lambda_1$ and $\lambda_2$ are the tuning parameters with a non-negative value.

The tuning parameters control the complexity of the estimate model, which is roughly represented by the effective number of parameters in the model. When $\lambda_1 = \lambda_2 = \infty$, $\hat{\boldsymbol{\beta}}_{\lambda_1, \lambda_2} = 0$. Especially, when $\lambda_1 = 0$, $\lambda_2 = \infty$, $\hat{\boldsymbol{\beta}}_{\lambda_1, \lambda_2}$ is given by the minimizer of $L(\boldsymbol{\beta})$ with constraints $\beta_j = \beta_{j'}$ for all $1 \leq j, j' \leq p$. When $\lambda_1 = 0$, $\lambda_2 = 0$, $\hat{\boldsymbol{\beta}}_{\lambda_1, \lambda_2}$ is the minimizer of $L(\boldsymbol{\beta})$ without the constraints. For $\lambda_1 = \lambda_2$, the proposed estimator is a minimizer of the empirical risk function with constraints:

$$\hat{\boldsymbol{\beta}} = \mathrm{argmin}_\beta\, L(\boldsymbol{\beta}), \tag{4}$$

subject to $\sum_{j=1}^p |\beta_j| + \sum_{(j,k)\in N} |\beta_j - \beta_k| \leq C$ for some $C$, which is a non-negative constant corresponding to $\lambda_1$. Figure 1 shows the boundaries of regions created by the constraints of lasso and fused lasso penalty function. The non-differentiable points on the boundary give a sparse or grouped solution minimizing $L(\boldsymbol{\beta})$.

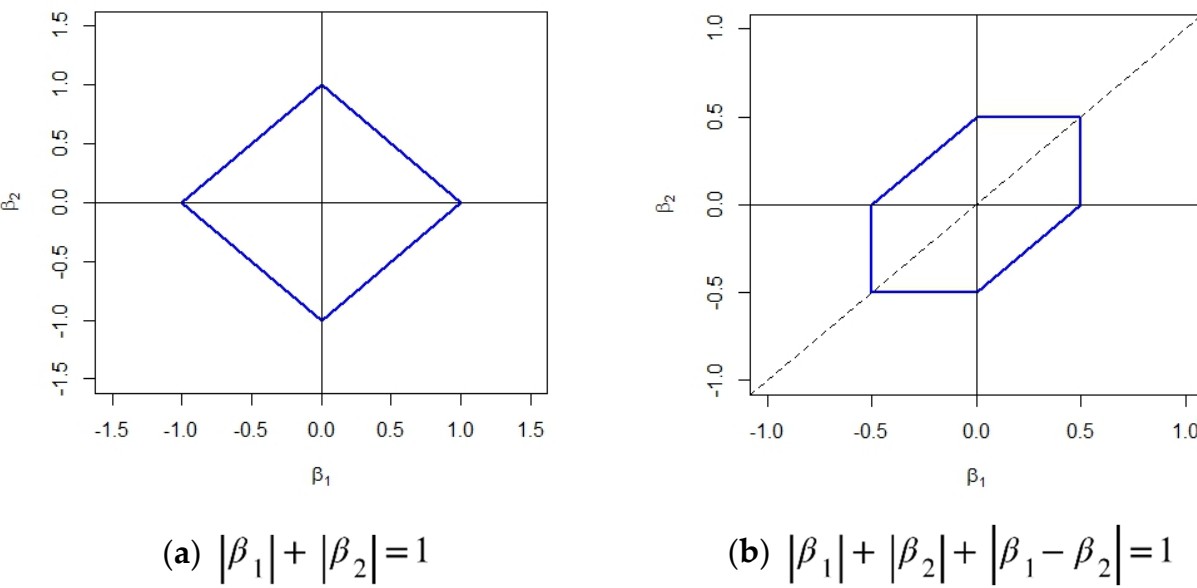

**(a)** $|\beta_1| + |\beta_2| = 1$　　　　　　　**(b)** $|\beta_1| + |\beta_2| + |\beta_1 - \beta_2| = 1$

**Figure 1.** Boundary regions of the constraints: lasso (**a**) and fused lasso (**b**).

**Remark 2.** *The network $N \subset \{(j,k) : 1 \le j, k \le p\}$ in the penalty function $\mathrm{p}_\lambda(\boldsymbol{\beta})$ can be chosen adaptively by observations. If an initial estimator $\widetilde{\beta}_j$ for $\beta_j$ with asymptotic variance being $s_j^2$ is available, then we can use $N = \left\{ (j,k) : \frac{|\widetilde{\beta}_j - \widetilde{\beta}_k|}{\sqrt{s_j^2 + s_k^2}} \le C, \ 1 \le j, k \le p \right\}$ for some C. The edge in the network N is based on the test statistics for the regression coefficients from two independent populations. That is, for the pairs with significantly different regression coefficients, the associated parameters are not regularized for grouping.*

*3.2. Recovering Procedure*

The estimator $\hat{\boldsymbol{\beta}}_{\lambda_1, \lambda_2}$ is known as a biased estimator, which shrinks toward zero or grouped coefficients. Even when the coefficients are well clustered, the bias can mislead the estimate of the linear trends. To avoid this problem, a recovering procedure is considered, which refits the linear trends on the clustered sites. Recall that $\boldsymbol{U}(\boldsymbol{\beta})$ is a set of unique elements of $\boldsymbol{\beta}$, and $\boldsymbol{G}(\boldsymbol{\beta})$ is the set of indices of grouped variables. With $\hat{\boldsymbol{\beta}}_{\lambda_1, \lambda_2}$, homogeneous sites are given by $\boldsymbol{G}\left(\hat{\boldsymbol{\beta}}_{\lambda_1, \lambda_2}\right)$. Then, a linear trend in a homogeneous cluster is estimated by

$$\hat{\beta}^{rc}{}_{h, \lambda_1, \lambda_2} = \mathrm{argmin}_\gamma \sum_{j \in G_k(\hat{\beta}_{\lambda_1, \lambda_2})} \sum_{t, t' \in Y_j} \left| \widetilde{u}_{tt'j} - (t - t')\gamma \right|, \tag{5}$$

For $h \in G_k\left(\hat{\boldsymbol{\beta}}_{\lambda_1, \lambda_2}\right)$ and $k = 1, \dots, \left| \boldsymbol{U}\left(\hat{\boldsymbol{\beta}}_{\lambda_1, \lambda_2}\right) \right|$. We call $\hat{\boldsymbol{\beta}}^{rc}{}_{\lambda_1, \lambda_2}$ a recovered estimator for linear trends. Then, a set of recovered parameters is constructed by varying the regularization parameters $\lambda_1$ and $\lambda_2$, denoted by $\mathbf{B} = \left\{ \hat{\boldsymbol{\beta}}^{rc}{}_{\lambda_1, \lambda_2} \in R^p : 0 \le \lambda_1, \lambda_2 \le \infty \right\}$, and the best estimate is chosen according to a model selection criteria.

*3.3. Model Selection*

The tuning parameters $\lambda_1$ and $\lambda_2$ control a complexity of the estimated model. The desired asymptotic properties, such as selection consistency, risk optimality, always require a proper selection of the tuning parameters in the regularization method. Under too large $\lambda_1$ and $\lambda_2$, too simple a model is obtained, where the unique groups of estimated coefficients are not separated.

Under too small $\lambda_1$ and $\lambda_2$, the grouping of homogeneous sites fails. Generally, the Bayesian information criterion (BIC) [25] is widely used to choose the tuning parameters. In the quantile regression, BIC is derived by Ref [29], which showed the robustness and model selection consistency of the BIC. We modify the conventional BIC [29] in the proposed regression model because the response variables are highly correlated. The BIC of the estimated model through $\hat{\boldsymbol{\beta}}^{rc}{}_{\lambda_1, \lambda_2}$ is defined by

$$\mathrm{BIC}(\lambda_1, \lambda_2) = \log\left( \sum_{i=1}^{n} \left| y_i - x_i^T \hat{\beta}^{rc}{}_{\lambda_1, \lambda_2} \right| \right) + \left| \boldsymbol{U}(\hat{\beta}^{rc}{}_{\lambda_1, \lambda_2}) \right| \frac{\log(n^*)}{2n^*}, \tag{6}$$

where $\left| \boldsymbol{U}(\hat{\beta}^{rc}{}_{\lambda_1, \lambda_2}) \right|$ is the number of unique elements in $\boldsymbol{U}(\hat{\beta}^{rc}{}_{\lambda_1, \lambda_2})$, and $n^*$ is the number of observations. In the proposed regression model, $y_\mathrm{i}$ for $i = 1, \dots, n$, is a pairwise difference of two response variables. For a site $j$, the number of $\{y_i\}$ corresponding to the site $j$ is the number of elements in $\widetilde{Y}_j$ such that the value of $\sum_{i=1}^{n} \left| y_i - x_i^T \hat{\boldsymbol{\beta}}^{rc}{}_{\lambda_1, \lambda_2} \right|$ has too much influence on BIC. For this reason, correcting the $n$ by $n^*$ can be understood as employing the effective number of observations. By choosing $\lambda_1$ and $\lambda_2$, the model minimizing $\mathrm{BIC}(\lambda_1, \lambda_2)$ on $\mathbf{B} = \left\{ \hat{\boldsymbol{\beta}}^{rc}{}_{\lambda_1, \lambda_2} \in R^p : 0 \le \lambda_1, \lambda_2 \le \infty \right\}$ is selected.

### 3.4. Optimization

The proposed estimator is the minimizer of the convex function with non-differentiable points. Generally, the optimization of the non-differentiable function requires much more computational cost than the differentiable function. In particular, the fused lasso penalty increases computational complexity in obtaining $\hat{\beta}_{\lambda_1, \lambda_2}$. For example, the coordinate algorithm [30] fails to achieve the minimum of the objective function due to non-differentiability of $\sum_{(j,k) \in N} |\beta_j - \beta_k|$ at $\beta_j = \beta_k$ for $(j, k) \in N$. It is well known that the quantile regression can be solved by linear programing, which is an optimization algorithm of a linear objective function, subject to linear equality constraints. Thus, the objective function and its constraint are given as follows:

$$\min \sum_{i=1}^{n} (\zeta_i^+ + \zeta_i^-) + \lambda_1 \sum_{j=1}^{p} \left( \beta_j^+ + \beta_j^- \right) + \lambda_2 \sum_{(j,k) \in N} \left( \beta_{jk}^+ + \beta_{jk}^- \right) \tag{7}$$

$$\text{s.t. } \zeta_i^+ - \zeta_i^- = y_i - x_i^T \beta, \ \zeta_i^+, \zeta_i^- \geq 0 \ (i = 1, \ldots, n)$$

$$\beta_j = \beta_j^+ - \beta_j^-, \ \beta_j^+, \beta_j^- \geq 0 \ (j = 1, \ldots, p)$$

$$\beta_{jk}^+ - \beta_{jk}^- = \beta_j - \beta_k, \ \beta_{jk}^+, \beta_{jk}^- \geq 0 \ (j, k) \in N.$$

Equation (7) is the minimization problem with respect to $\zeta_i^+$, $\zeta_i^-$ for $i = 1, \ldots, n$, $\beta_j^+$, $\beta_j^-$ for $j = 1, \ldots, p$ and $\beta_{jk}^+$, $\beta_{jk}^-$ for $(j, k) \in N$. Then, the solution of $\beta = (\beta_1, \ldots, \beta_p)^T$ in Equation (3) is given by $\beta_j = \beta_j^+ - \beta_j^-$ for $j = 1, \ldots, p$, where $\beta_j^+$ and $\beta_j^-$ are the solution of Equation (7). We implement the optimization algorithm using R, whose code is found in https://github.com/chulhongsung/TrendClustering (accessed on 24 May 2021).

**Remark 3.** *When n is large, the computational problem frequently occurs in maintaining physical memory on the linear programing. In this case, we can replace the loss function in Equation (3) by Huber loss [31] and apply the alternative directional multiplier method (ADMM) [32] to minimize Equation (3). The ADMM consists of two convex problems, which can be easily solved by the MM algorithm [33]. The computational cost of the ADMM does not depend much on n, such that we can avoid the scalable problem in memory. However, the ADMM generally requires a large number of iterations, so we should consider the time cost before applying the ADMM.*

## 4. Numerical Studies

### 4.1. Simulation

We compare the performance of the proposed method with those of the other four methods: the naive LSE(nLSE), the naive LAD(nLAD) estimator, the rank LSE(rLSE), the rank LAD(rLAD) estimator. The first two methods use the conventional linear regression model with $u_{tj}$, as the response variable and estimate coefficients by minimizing the empirical risk with $l_2$ and $l_1$ loss function, respectively. The latter two methods use the differences of the observed value $u_{tj} - u_{t'j}$ and estimate the coefficients with $l_2$ and $l_1$ loss function, which is known as rank regression [21]. These methods are known as being robust to the underlying distributions of $u_{tj}$, while all methods except the proposed one do not provide grouped coefficients. $u_{tj}$ is generated from GEV($\xi_j + \beta_j t$, $\alpha_j$, $\kappa_j$) for $t = 1, \ldots, T$ and $j = 1, \ldots, p$, and the heterogeneity of characteristics of sites is modeled by $\xi_j \sim N(120, 20)$, $\alpha_j \sim N(40, 10)$, $\kappa_j \sim U(-0, 3, 0.3)$. $a_r$ denotes $(a, \ldots, a) \in R^r$. Throughout all simulations, let $p = 30$ and $T = 30$, and the performances of the considered estimators are measured by 200 repetitions. In simulation 1, let $\beta = (-5_{10}, 0_{10}, 5_{10})$ and $N = \{(j, k) : 0 < |j - k| \leq 15\}$. In this case, the linear time trends in the sites are clustered into three large groups. Figure 2 shows the mean squared error of differences between the estimated parameters and the true parameters.

The proposed method is the best, and the naive LSE, or the rank LSE, is worst (note that naive LSE or rank LSE are exactly equal in linear time trend model). It is found that the naive LAD is slightly better than the naive LSE. Using $l_1$ loss function, the variances are reduced due to the robustness of the LAD estimator to heavy tailed distribution. The rank LAD has smaller MSE than the naive LAD. The proposed method shows critical improvements in estimating the coefficients compared with the rank LAD. Since the proposed method gives grouped estimates, the variances are reduced, such that the MSE, surprisingly, decreases, as shown in the left panel of Figure 3. In this simulation, the proposed method gives 3.6 clusters of estimated coefficients, on average. In simulation 2, we let $\beta = (-5_5, -2.5_5, 0_{10}, 2.5_5, 5_5)$ and $N = \{(j, k) : 0 < |j - k| \leq 15\}$. In this case, the strength of signal in the time trend is weaker, and the grouping coefficient is more difficult than in simulation 1. As shown in Figure 4, in MSE, the proposed method is still the best, and the naive LSE is the worst. In this simulation, the rank LAD estimator is still better than the naive LAD estimator. However, the improvement of the proposed method becomes less compared with the result in simulation 1. Figure 5 shows that a promising improvement of the proposed method is not achieved when the true coefficients between clusters are not separated well.

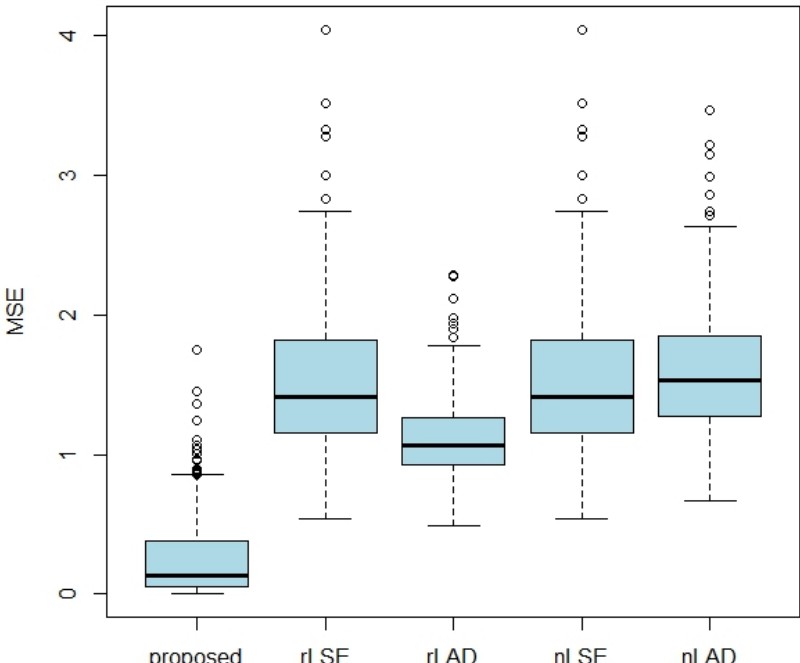

**Figure 2.** MSE of estimated coefficients of five methods in simulation 1. The five methods are our proposed method(proposed), the naive LSE(nLSE), the naive LAD(nLAD), the rank LSE(rLSE) and the rank LAD(rLAD).

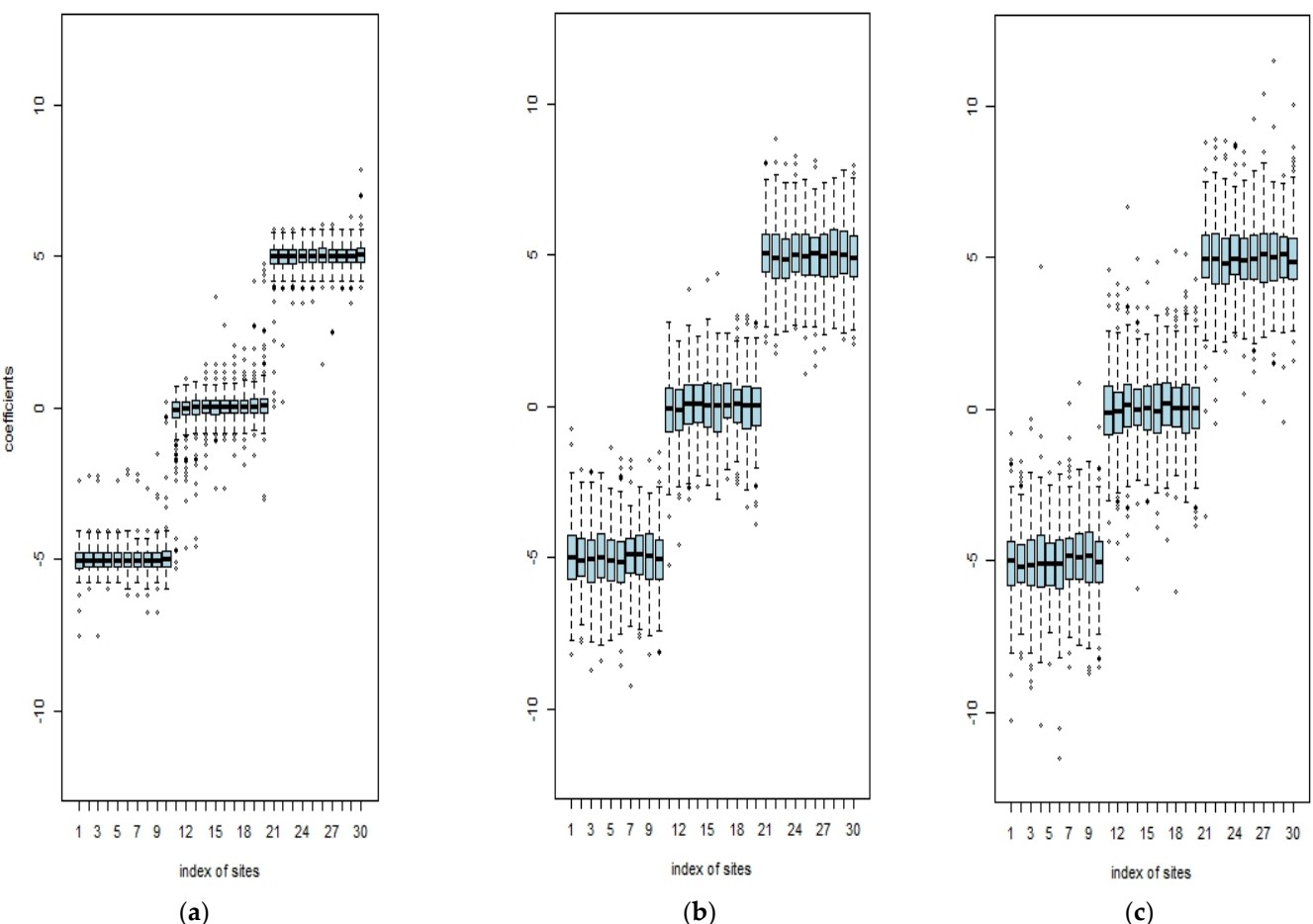

**Figure 3.** Estimated coefficients of the proposed method (**a**), rLSE (**b**) and rLAD (**c**) in simulation 1.

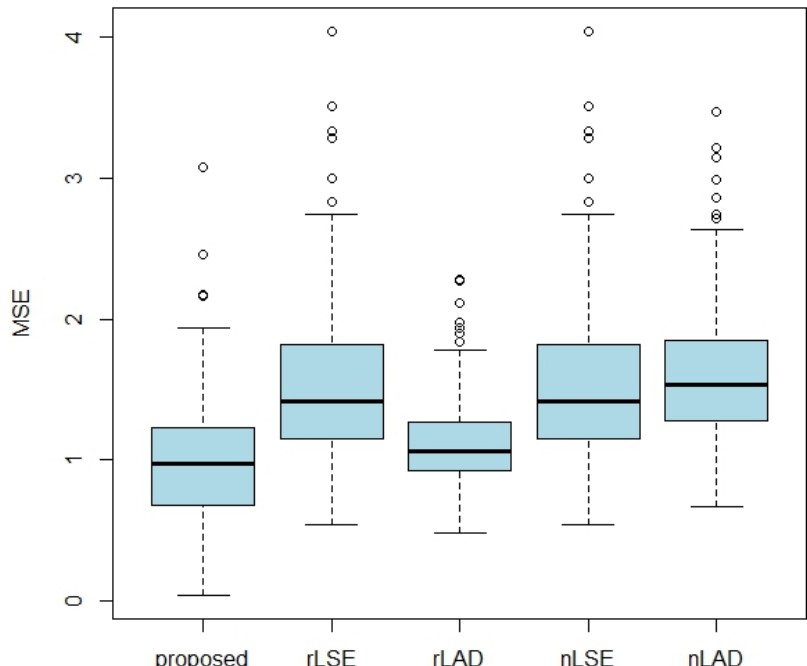

**Figure 4.** MSE of estimated coefficients of five methods in simulation 2. The five methods are our proposed method(proposed), the naive LSE(nLSE), the naive LAD(nLAD), the rank LSE(rLSE) and the rank LAD(rLAD).

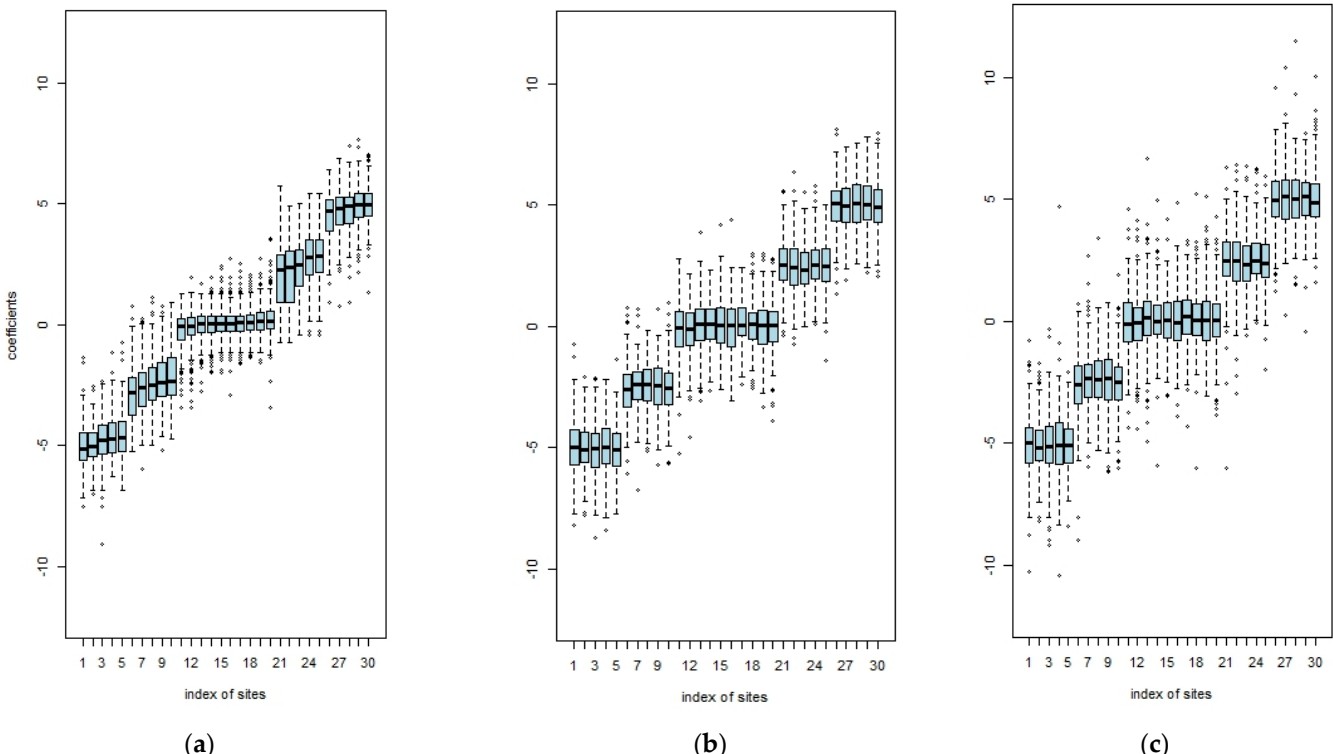

**Figure 5.** Estimated coefficients of the proposed method (**a**), rLSE (**b**) and rLAD (**c**) in simulation 2.

*4.2. Real Data Analysis*

We apply the proposed method to estimate the linear time trend of the annual maximum of daily precipitation in South Korea. Our research area, the southern portion of the Korean Peninsula, is located in east Asia between the longitude 126°–132° and latitude 33°–38°. About 70% of the Korean Peninsula's topography is composed of mountains. One of the important geographical factors affecting the climate in South Korea, the Taebaek Mountain range, is located on the east side from north to south. It creates overarching geographical property, which is east high and west low and climate diversity of South Korea (Figure 6). This area has four seasons, with glaringly distinctive climate features. In particular, the amount of precipitation is mainly in the summer, and 50 to 60% of the annual precipitation is concentrated at this time. The annual maximum of daily precipitation from 1971 to 2021 collected from 60 meteorological observatories is analyzed, and the information about the observatories is listed in Table A1. Define a set of indices of site pairs by neighborhood $N$ defined in Section 3.1, whose distance is less than 100 km. In addition, let $\widetilde{Y_j}$ by $\left\{ (t, t') : t, t' \in Y_j, \, 0 < |t - t'| \leq 50 \right\}$. By varying the tuning parameter, the set of estimates is obtained, and the tuning parameter, the minimizing BIC in Section 3.3, is chosen. The BIC is minimized at $\lambda_1 = \lambda_2$ and $\lambda_2 = 330$. Figure 7 illustrates the solution sets defined by $\left\{ \boldsymbol{\beta} : \boldsymbol{\beta} = \hat{\boldsymbol{\beta}}^{rc}_{\lambda_1, \lambda_2}, \, \lambda_1 = \lambda_2 \text{ and } \lambda_2 > 0 \right\}$ and shows the BIC corresponding to the solution sets. The estimated trend coefficients achieving the minimum BIC are provided in Table A2.

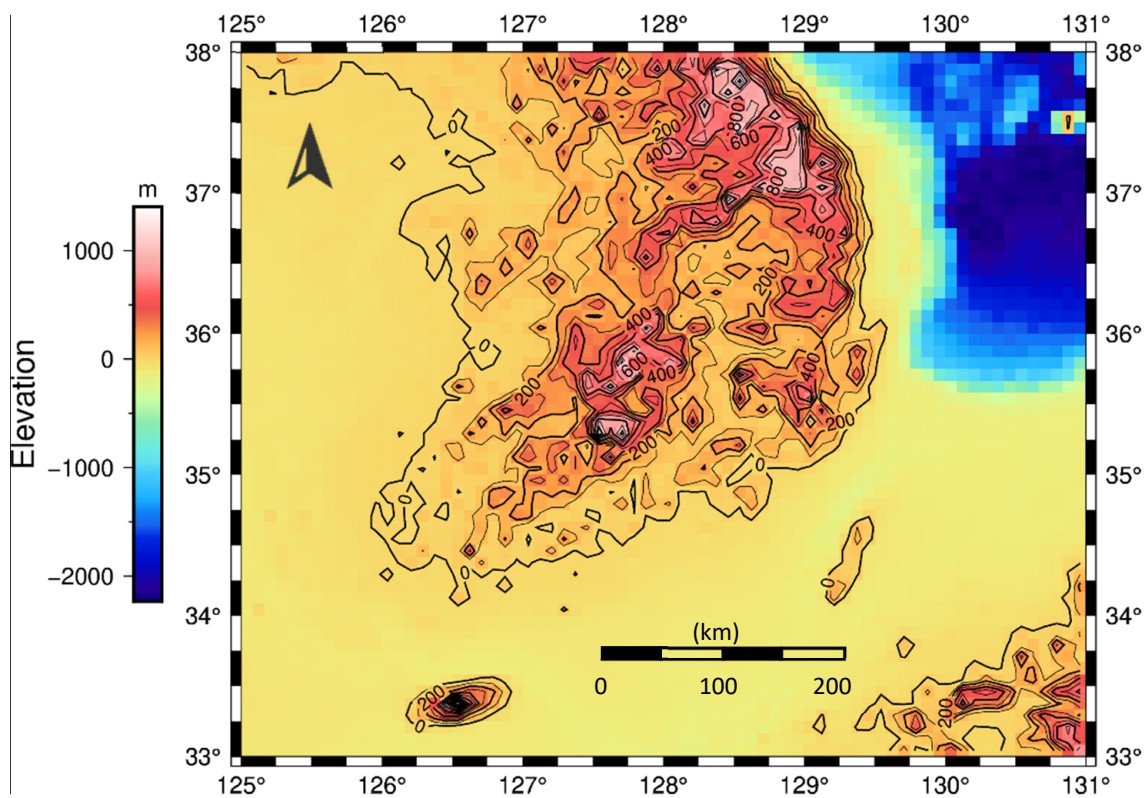

**Figure 6.** Southern portion of the Korean Peninsula and islands and their elevation (m).

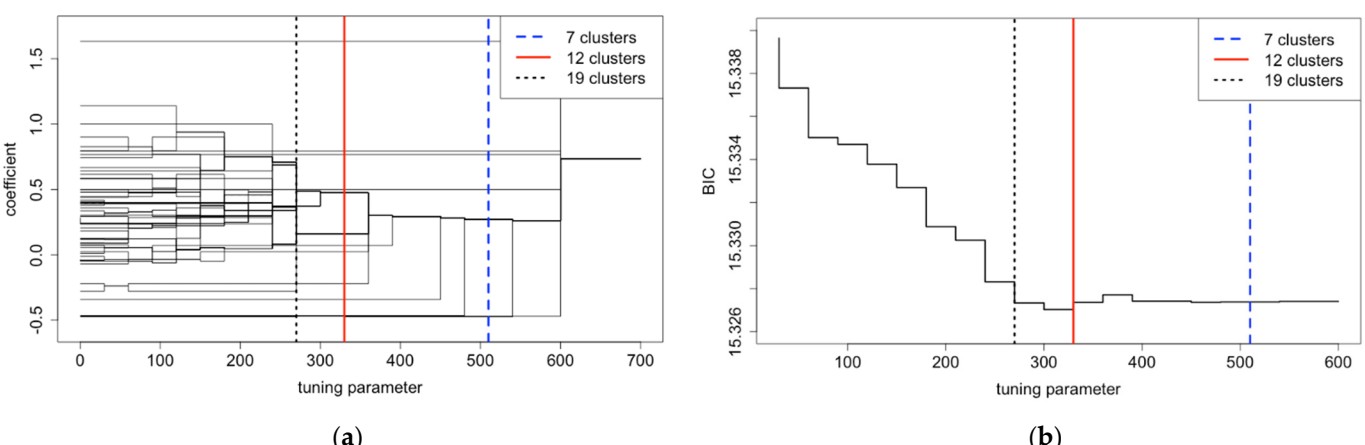

(**a**)  (**b**)

**Figure 7.** Path of coefficients (**a**) and BIC plot (**b**) with respect to tuning parameter $\lambda_2$.

The coefficients are clustered into twelve groups. The largest group consists of 27 sites, which are located in southeastern coastal line of South Korea. The estimated trend coefficient of the group is about 0.47, such that the increasing trend of extreme rainfall is detected. The second largest group consists of 22 sites, which are located in the midland, and the estimates are about 0.16. From the obtained estimates, the trends in the west and the midland are much weaker than the ones in the south and east. In the western coastal site, the homogenous site is estimated, and relatively weak trend is detected.

Figure 8 summarizes the results that display the trend coefficient on the southern portion of the Korean Peninsula by the Kriging method with Gaussian filter. The trend coefficients illustrated in Figure 8 are estimated with $\lambda_1 = \lambda_2$ and $\lambda_2 = 510, 330, 270$. Even though the trend maps are different to each other, it is remarkable that the pattern in the increasing trend is separated in western sites and southern sites when the number

of clusters grows. Moreover, Figure 8 indicates that weak trends in the western coastal line and the midland are estimated, and strong positive trends in Jeju, which is the largest island in South Korea, and Ulleung, which is an island located 120 km east of the Korean Peninsula, are estimated. As the clusters are subdivided, the negative trend in Ganghwa, which is located in the northwestern end, tends to be clear. Compared to the west site, the east site has a relatively strong and nonstationary trend in heavy rainfall. These results imply that (1) the intensity of heavy rainfall tends to be increasing, and nonstationarity, represented by trend coefficients of the Jeju and Ulleung islands, is more severe; (2) the heavy rainfall trend in the coastal site has distinct characteristics depending on its location; (3) the trend in Ganghwa uniquely shows a negative trend.

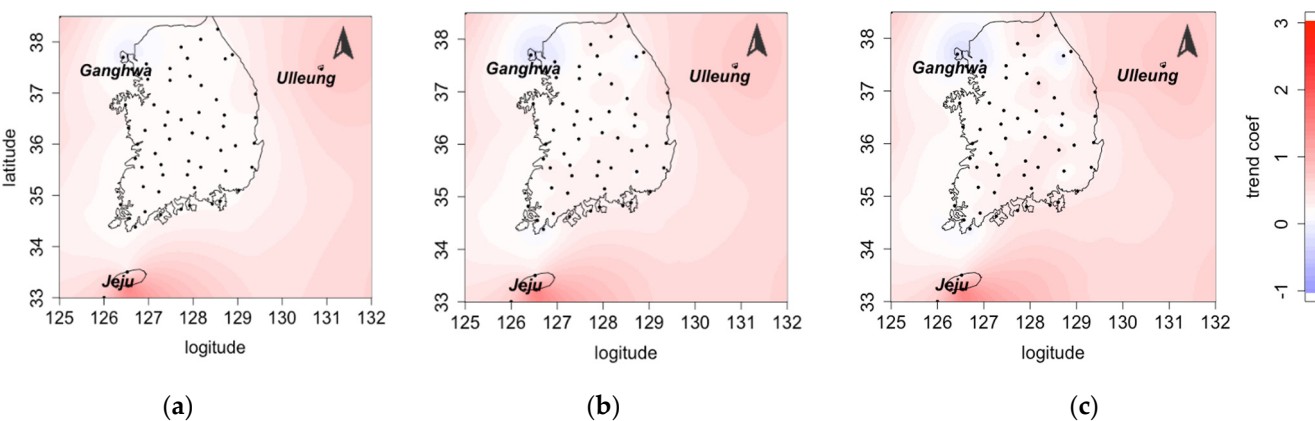

**Figure 8.** Trend map of southern portion of the Korean Peninsula with 7 clusters (**a**), 12 clusters (minimum BIC) (**b**) and 19 clusters (**c**).

## 5. Discussion and Concluding Remarks

We proposed a new method of simultaneously detecting and clustering regional time trends in the nonstationary distribution. The proposed method has three advantages. The first is that it does not depend on the underlying distribution of the observations. When the de-trending method is considered in the conventional RFA, it is essential to estimate the linear trend without any knowledge of the unknown distribution. The proposed method is distribution free, so the moment condition corresponding to the underlying distribution is not required. The second is that the proposed method provides an easy way to determine the number of clusters. Even though the typical clustering algorithm can be applied to group the statistics for the linear trends, it is not clear how to consider the estimation errors in the clustering algorithm. However, since our method gives clustered coefficients based on the empirical risk, the information criteria guarantee the selection of optimal clusters, theoretically. Lastly, the variance of estimated linear time trends is reduced. By pooling observations [21], more accurate estimated trends can be obtained.

In the real data analysis for the annual maximum daily precipitation, it is found that there is a pattern of homogeneity of nonstationarity across the South Korean peninsula. The result of our analysis shows that there are different patterns in the south and east sites and the west sites of the South Korean peninsula. The decreasing or stationary trends of extreme precipitation were found in the western sites. On the contrary, it was found that positive linear trends are clustered in the sites of eastern and southern coastal lines, and it is expected that the extreme events are more frequently observed on those sites. These results are interesting, in that Ref [11] found that the probability of extreme events was higher in the southern sites than others; however, our result shows that temporal trends of extreme events are decreasing in the southern sites. This means that a further analysis should be performed to detect the patterns of extreme climate events. We believe that our work can contribute to future research on regional frequency analysis under nonstationary distribution.

However, the proposed estimator severely depends on the tuning parameter selection. For example, a careless selection of the tuning parameter can mislead the result of estimating

the linear time trend. Thus, it is essential to develop or justify a tuning parameter selection method, and we leave this problem for our future work.

**Author Contributions:** Conceptualization, J.-J.J., Y.K.; methodology, S.H., J.-J.J.; writing—review and editing, S.H., J.-J.J., Y.K.; visualization, S.H.; supervision, Y.K. All authors have read and agreed to the published version of the manuscript.

**Funding:** Jeon was supported by the 2018 Research Fund of the University of Seoul.

**Institutional Review Board Statement:** Not applicable.

**Informed Consent Statement:** Not applicable.

**Data Availability Statement:** The data are available at https://data.kma.go.kr/data/grnd/selectAsosRltmList.do (accessed on 18 May 2022).

**Conflicts of Interest:** The authors declare no conflict of interest.

## Appendix A

**Table A1.** Site information.

| Site ID | Site | Latitude | Longitude | Site ID | Site | Latitude | Longitude |
|---|---|---|---|---|---|---|---|
| 90 | Sokcho | 38°15′ | 128°33′ | 202 | Yangpyeong | 37°29′ | 127°29′ |
| 100 | Daegwallyeong | 37°40′ | 128°43′ | 203 | Icheon | 37°15′ | 127°29′ |
| 101 | Chuncheon | 37°54′ | 127°44′ | 211 | Inje | 38°03′ | 128°10′ |
| 105 | Gangneung | 37°45′ | 128°53′ | 212 | Hongcheon | 37°41′ | 127°52′ |
| 108 | Seoul | 37°34′ | 126°57′ | 221 | Jecheon | 37°09′ | 128°11′ |
| 112 | Incheon | 37°28′ | 126°37′ | 226 | Boeun | 36°29′ | 127°44′ |
| 114 | Wonju | 37°20′ | 127°56′ | 232 | Cheonan | 36°46′ | 127°07′ |
| 115 | Ulleung | 37°28′ | 130°53′ | 235 | Boryeong | 36°19′ | 126°33′ |
| 119 | Suwon | 37°16′ | 126°59′ | 236 | Buyeo | 36°16′ | 126°55′ |
| 129 | Seosan | 36°46′ | 126°29′ | 238 | Geumsan | 36°06′ | 127°28′ |
| 130 | Uljin | 36°59′ | 129°24′ | 243 | Buan | 35°43′ | 126°42′ |
| 131 | Cheongju | 36°38′ | 127°26′ | 244 | Imsil | 35°36′ | 127°17′ |
| 133 | Daejeon | 36°22′ | 127°22′ | 245 | Jeongeup | 35°33′ | 126°51′ |
| 135 | Chupungnyeong | 36°13′ | 127°59′ | 247 | Namwon | 35°24′ | 127°19′ |
| 136 | Andong | 36°34′ | 128°42′ | 256 | Juam | 35°04′ | 127°14′ |
| 138 | Pohang | 36°01′ | 129°22′ | 260 | Jangheung | 34°41′ | 126°55′ |
| 140 | Gunsan | 36°00′ | 126°45′ | 261 | Haenam | 34°33′ | 126°34′ |
| 143 | Daegu | 35°53′ | 128°37′ | 262 | Goheung | 34°37′ | 127°16′ |
| 146 | Jeonju | 35°49′ | 127°09′ | 272 | Yeongju | 36°52′ | 128°31′ |
| 152 | Ulsan | 35°33′ | 129°19′ | 273 | Mungyeong | 36°37′ | 128°08′ |
| 156 | Gwangju | 35°10′ | 126°53′ | 277 | Yeongdeok | 36°31′ | 129°24′ |
| 159 | Busan | 35°06′ | 129°01′ | 278 | Uiseong | 36°21′ | 128°41′ |
| 162 | Tongyeong | 34°50′ | 128°26′ | 279 | Gumi | 36°07′ | 128°19′ |
| 165 | Mokpo | 34°49′ | 126°22′ | 281 | Yeongcheon | 35°58′ | 128°57′ |
| 168 | Yeosu | 34°44′ | 127°44′ | 284 | Geochang | 35°40′ | 127°54′ |
| 170 | Wando | 34°23′ | 126°42′ | 285 | Hapcheon | 35°33′ | 128°10′ |
| 184 | Jeju | 33°30′ | 126°31′ | 288 | Miryang | 35°29′ | 128°44′ |
| 189 | Seogwipo | 33°14′ | 126°33′ | 289 | Sancheong | 35°24′ | 127°52′ |
| 192 | Jinju | 35°09′ | 128°02′ | 294 | Geoje | 34°53′ | 128°36′ |
| 201 | Ganghwa | 37°42′ | 126°26′ | 295 | Namhae | 34°48′ | 127°55′ |

**Table A2.** Trend coefficients of sites.

| Site ID | Site | Trend Coef. | Site ID | Site | Trend Coef. |
|---|---|---|---|---|---|
| 90 | Sokcho | 0.474071 | 202 | Yangpyeong | 0.159259 |
| 100 | Daegwallyeong | −0.465909 | 203 | Icheon | 0.159259 |
| 101 | Chuncheon | 0.474071 | 211 | Inje | 0.474071 |
| 105 | Gangneung | 0.474071 | 212 | Hongcheon | 0.159259 |
| 108 | Seoul | −0.465909 | 221 | Jecheon | 0.474071 |
| 112 | Incheon | 0.070833 | 226 | Boeun | 0.159259 |
| 114 | Wonju | 0.159259 | 232 | Cheonan | 0.159259 |
| 115 | Ulleung | 0.766667 | 235 | Boryeong | 0.159259 |
| 119 | Suwon | 0.159259 | 236 | Buyeo | 0.159259 |
| 129 | Seosan | 0.159259 | 238 | Geumsan | 0.159259 |
| 130 | Uljin | 0.793181 | 243 | Buan | 0.159259 |
| 131 | Cheongju | 0.159259 | 244 | Imsil | 0.159259 |
| 133 | Daejeon | 0.159259 | 245 | Jeongeup | 0.474071 |
| 135 | Chupungnyeong | 0.159259 | 247 | Namwon | 0.474071 |
| 136 | Andong | 0.474071 | 256 | Juam | 0.474071 |
| 138 | Pohang | 0.474071 | 260 | Jangheung | 0.474071 |
| 140 | Gunsan | 0.159259 | 261 | Haenam | 0.159259 |
| 143 | Daegu | 0.474071 | 262 | Goheung | 0.474071 |
| 146 | Jeonju | 0.159259 | 272 | Yeongju | 0.159259 |
| 152 | Ulsan | 0.474071 | 273 | Mungyeong | 0.159259 |
| 156 | Gwangju | 0.159259 | 277 | Yeongdeok | 0.474071 |
| 159 | Busan | 0.474071 | 278 | Uiseong | −0.220588 |
| 162 | Tongyeong | 0.474071 | 279 | Gumi | 0.474071 |
| 165 | Mokpo | 0.023529 | 281 | Yeongcheon | 0.474071 |
| 168 | Yeosu | 0.474071 | 284 | Geochang | 0.474071 |
| 170 | Wando | −0.341463 | 285 | Hapcheon | 0.474071 |
| 184 | Jeju | 0.5 | 288 | Miryang | 0.474071 |
| 189 | Seogwipo | 1.634286 | 289 | Sancheong | 0.474071 |
| 192 | Jinju | 0.474071 | 294 | Geoje | 0.474071 |
| 201 | Ganghwa | −0.47 | 295 | Namhae | 0.474071 |

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
