# Peer review of "Clustering for Regional Time Trend in the Nonstationary Extreme Distribution"

_water, doi:10.3390/w14111720_

Round 1

Reviewer 1 Report

In the reviewed manuscript, there are several issues that need to be resolved, including:

- clearly indicate the purpose of the research,

- provide a short description of the research area

- provide whether the proposed method has been verified on independent material

- the areas indicated in the manuscript (eg. Page 9 line 294, Page 10 line 295-297) should be shown in Figure, which will make reading easier for the reader

- the legend and axis titles are not legible

- point 5 - in the opinion of the reviewer, the discussion is too general,

- there is lack of "conclusion" - it is necessary to provide conclusions from the research

-  indicate the statistical significance of the trend (page 11 line 336 - Table A2)

Detailed comments:

Page 1 Abstract

The abstract should be complemented with research results

Page 1 line 22 point 1

"In extreme data analysis with hydrometeorological variables, the number of available observations is generally small" - the notation is not clear. Literature should be also referred to

Page 2 line 56 point 1

I suggest clearly indicating the purpose of the research

Page 6 line 208 point 3.4

"For example, the coordinate algorithm (Friedman et al., 2010) ..." - I suggest writing citations equally throughout the manuscript, in accordance with the journal's requirements

Page 9 line 281 point 4.2

“The coefficients are clustered into twelve groups. The largest group consists of many south-eastern sites within coastal line in the South Korea. " - the notation is not clear

Page 9 line 272 , 274 point 4.2

"... Section 3.1 ..."; "... section 3.3." - I suggest writing equally (small / capital letter) throughout the manuscript

Page 9, 10 Figure 6, Figure 7

Legend and axis titles are not legible

Author Response

Thank you for your helpful comments. We tried to fully respond to your comments point by point and to show the purpose of our paper more clearly.

Reviewer 2 Report

Overall, the manuscript is well-written and the results are well-presented, except for the introduction section, which in my opinion is not complete.

The context of research should be well established by summarizing current understanding and background information about various available clustering techniques. I suggest in L32  briefly explain different clustering techniques. Also, the gaps and motivation of the work are not presented well. I suggest adding a paragraph to explain the motivation of the study.

I just have some more minor comments that need to be addressed before the paper is ready for publication.

L42-44. “For example, L-moments, a fundamental…” this sentence is not clear. Please revise and add a reference for the L-moment technique. See (Hosking and Wallis 1993) and (Hosking and Wallis 1997)

x-labels and y-labels are too small in all figures specifically figures 6 and 7.

Figure 2,4 caption. Abbreviations should be spelled out in captions.

References

Hosking, J. R. M., and Wallis, J. R. (1993). “Some Statistics Useful in Regional Frequency Analysis.” Water Resources Research, 29(92).

Hosking, J. R. M., and Wallis, J. R. (1997). Regional Frequency Analysis: An approach based on L-moments.

Author Response

(The authors gave the same response as above.)

Round 2

Reviewer 1 Report

Thank Authors of the manuscript for their answers;

The article has been greatly improved with some of the comments.

However, I still suggest complementing the manuscript with

- provide a short description of the research area -  precipitation included in the manuscript depends on geographical factors, including orographic, so it is necessary to provide a short description of the area

In the manuscript there is point 5 "Discussion and concluding remarks", there is no point "conclusion" - I suggest complementing and providing conclusions from the research

"The result of our analysis shows that there are different patterns in the south and east sites and the west site of the South Korean peninsula" - I suggest providing more detailed conclusions

Page 10 line 300: “Moreover, Figure 8 indicates that weak trends in western coastal line and midland are estimated and strong positive trends in Jeju which is the largest island in South Korea and Ulleung which is the island located 120 km east of the Korean Peninsula are estimated ”- I suggest including Figure 8. It is necessary to check the numbers of the figures and their quotation in the text

Author Response

Thank you for your kind comments. Please see the attached file.

Thank you!
